# Stabilization of Essential Oil: Polysaccharide-Based Drug Delivery System with Plant-like Structure Based on Biomimetic Concept

**DOI:** 10.3390/polym15163338

**Published:** 2023-08-08

**Authors:** Xue-Yee Lim, Jing Li, Hong-Mei Yin, Mu He, Ling Li, Tong Zhang

**Affiliations:** 1School of Pharmacy, Shanghai University of Traditional Chinese Medicine, Shanghai 201203, China; limxy0717@icloud.com (X.-Y.L.); lj2657350646@163.com (J.L.); 2Jiangsu Kanion Pharmaceuticals Co., Ltd., Lianyungang 222001, China; yinhongmei1010@126.com; 3School of Acupuncture-Moxibustion and Tuina, Shanghai University of Traditional Chinese Medicine, Shanghai 201203, China; h961030686@163.com; 4College of Pharmacy, Anhui University of Chinese Medicine, Hefei 230012, China

**Keywords:** polysaccharides, essential oil, biomimetic, drug delivery system, stability, emulsion, encapsulation, solidification

## Abstract

Essential oils (EOs) have stability problems, including volatility, oxidation, photosensitivity, heat sensitivity, humidity sensitivity, pH sensitivity, and ion sensitivity. A drug delivery system is an effective way to stabilize EOs, especially due to the protective effect of polymeric drug carriers. Polysaccharides are frequently employed as drug carrier materials because they are highly safe, come in a variety of forms, and have plentiful sources. Interestingly, the EO drug delivery system is based on the biomimetic concept since it corresponds to the structure of plant tissue. In this paper, we associate the biomimetic plant-like structures of the EO drug delivery system with the natural forms of EO in plant tissues, and summarize the characteristics of polysaccharide-based drug carriers for EO protection. Thus, we highlight the research progress on polysaccharides and their modified materials, including gum arabic, starch, cellulose, chitosan, sodium alginate, pectin, and pullulan, and their use as biomimetic drug carriers for EO preparations due to their abilities and potential for EO protection.

## 1. Introduction

Essential oil (EO) is a mixture of volatile and hydrophobic liquid that can be distilled with water vapor [1] (p. 226). It is biologically and medicinally active. Its pharmacological effects include anti-inflammatory, antimicrobial, anticancer, antiaging, antinociceptive, and neuroprotective properties [2,3,4]. After their synthesis from plastids, EO naturally accumulates in secretory structures such as oil cells, oil canals, oil chambers, granular hair, etc. [5]. However, it is volatile and loses its active constituents after extraction due to environmental factors such as light, heat, humidity, oxygen, etc. Hence, it is difficult to ensure efficacy and long-term preservation, which creates a huge challenge for EO preparations and other applications. Li et al. [6] studied the process of EO preparation and found that in contrast to crushing, concentration, and drying operations, the extraction process resulted in heat transformation and accelerated the volatilization of components. In their study, they also stated that the volatile components could be retained more when the raw powder of the plant was directly used to replace EO in the preparation, indicating that the plant cell structures were conducive to the stability of the EO. In other words, the stabilization of EO naturally depends on encapsulation and other protective effects of plant cell structures. Gao Bo et al. [7] defined raw powder as a raw material in EO preparation using the “unification of drugs and excipients” method, and also found that Humagsolan tablets that contain Angelica EO and are prepared using the unification of drugs and an excipient method are more stable than Humagsolan tablets prepared using the inclusion and spraying methods, which use extracted EOs as raw materials. Basically, the authors proposed that pulverization could be used as a drug preparation process that preserves the integrity of the plant cell wall while completely encapsulating EO. Further, the period of expiration (t_0.9_) of Humagsolan tablets prepared with raw Angelica powders was 6.7 and 2.7 times longer than that of EO-spraying tablets and cyclodextrin inclusion, respectively, proving that the structures of plant cells are protective of EO.

In fact, EO possesses antioxidant activities after being extracted, which means that EO is able to fight against environmental factors by itself [8]; however, this ability is low without plant cell protection, causing EO to easily lose flavor and even become darker and resin-like in preservation. Hence, carrier technology is widely used to improve the stability of EO. Compared with antioxidants and packaging, etc., carrier technology is more flexible and efficient. EO carrier technology is a type of drug delivery system that uses different materials to improve the stability of EO [9]. The different types of EO drug carriers, such as microemulsion, microspheres, etc., are usually polymeric [10,11,12]. Polymer materials are macromolecular compounds chiefly bonded by a great number of repeat units, which can carry various drug molecules based on large modules and long chains or network structures [13,14] (pp. 1, 11). The EO is encapsulated in the polymeric carrier structure, so that the artificial carrier system replaces the EO’s natural state in plant cells and achieves stability of the EO in vitro in the form of preparations; herein, this is based on the biomimetic concept.

Polysaccharides are carbohydrate polymers made up of lots of monosaccharide units [15] (p. 97). Polysaccharides are safe, biodegradable, and have abundant natural sources [16]; thus, they have become the preferred polymeric materials for EO drug delivery systems. Most of the polysaccharide carrier materials are extracted from plants (except for chitosan, which is mainly derived from animal shells) and, therefore, the composition of EO carriers is similar to the natural state of EO. That is, the polysaccharide-based delivery systems have a good ability to encapsulate EO and an advantage in the controlled release of EO, as is the case in plant tissues. For example, the content of EO is high in the early stage of plant flowering in order to provide nutrients, while at the end of the flowering stage, the release of EO is high to attract insect pollination [17]. The polysaccharide carriers can also provide different release properties due to their designation as a drug delivery system, ensuring that the EO reaches its target cells stably.

In view of this, this paper classifies the current polysaccharide-based drug delivery systems for EO into three types: emulsification, encapsulation, and solidification. Additionally, this paper highlights the similarity between the drug delivery system and natural forms of EO and relates it to the biomimetic concept. The polysaccharide materials, including gum arabic, starch, cellulose, chitosan, sodium alginate, pectin, and pullulan, are classified according to their main application to EO stabilization. Further, the characteristics of each polysaccharide are pointed out in terms of improving EO stability for biomimetic drug delivery systems that rely on carrier technology. This paper thus aims to suggest stabilization strategies for EO and its preparations.

## 2. Drug Delivery System and Plant-Like Structure in Relation to the Biomimetic Concept

As shown in Figure 1, there are three forms of EO in plants [18] (pp. 25, 26, 39–41), including:Ground tissues, such as parenchymal cells, mesophyll cells, and suberized cells. In ground tissues, EOs are in the form of small droplets. The micromorphology of ground tissues demonstrates that the oil droplets are shielded by layers of cell walls because either cell grids containing several oil droplets in the tissue space or the oil droplets are surrounded by adjacent cells [19,20].External secretory tissues, such as glandular hairs and secretory epidermis. In external secretory tissues, the droplets of EO exist in the subcuticular storage cavities of secretory cells [21]. In parallel, there are non-cellulosic polysaccharides secreted with EO and transported to the cavity space [22]. Hence, the movement of EO molecules is limited by the heterogenous environment and cuticle barrier. There are various secretory structures in terms of micromorphology, but droplets of EO can be seen that were stored in cavities formed through isolation of the cuticular layer of the epidermis.Internal secretory tissues, such as the oil cell, oil chamber, vitta, resin canal, laticifer, etc. In internal secretory tissues, EO exists in the oil cavity bounded by vacuolization during secretion [23]. Hence, the oil drops have multiprotection. The multiprotection given to the oil drops is comprehensively provided by secretory cell walls, fluid of vacuoles, other secretions such as latex, adjacent secretory cells, and ground tissues.

Polysaccharides form drug carriers to encapsulate EOs for protection, like plant tissues, by encasing the cells inside them. Drug carrier technologies include emulsification, encapsulation, and solidification, which return the extracted EO to its natural microscopic droplet form. Carrier technologies include emulsification, encapsulation, and composite technology. The designing of EO drug carrier systems is commonly performed via simulation of the characteristics of plant tissue structures. Herein, there is a biomimetic concept involved. In brief, the EO drug carrier systems correspond to the three protective effects provided by three tissue structures and are based on the following biomimetic carrier technologies:Emulsification: biomimetic carrier technology based on the protection of heterogeneous dispersion.

Biomimetic carrier technology based on the protection of heterogeneous dispersion involves the mixing of oil, water, and amphiphilic molecules to form an emulsion, which simulates the external secretory tissues, and is widely known as emulsification technology. Emulsification technology relies on the orientational self-wedging behavior of polymer amphiphilic groups at the oil–water interface when mixing to reduce the interfacial tension [24,25]. This will incidentally form a great number of drug carriers named emulsion droplets, which become dispersed in the heterogenous medium. The emulsifying systems of EOs are always of the O/W type to allow the EOs to be loaded into the core of the emulsion droplets according to the solubility rule of “like dissolves like”. Hence, the EOs droplets are protected by an interfacial film and a heterogenous medium, that is, the dispersion of EO droplets in heterogenous medium and interfacial barrier protection may confine the EOs’ volatility and sensitivity to environment factors. In terms of micromorphology, the emulsion carrier system is characterized by spherical emulsion droplets with an oil core and a layer of polymeric interfacial film in continuous water phase.

From the perspective of chemical structures, polysaccharide molecules, which always have a great number of hydroxyl groups, are hydrophilic. They are also frequently connected to hydrophobic groups such as methyl groups, making them suitable for usage as a surfactant to stabilize the interface while mixing EO and water [26,27,28]. Both emulsion carrier systems and plant external secretory tissues use an interfacial membrane and a heterogenous medium as a protective barrier for EO. In conclusion, the emulsion carrier system is a biomimetic structure that simulates the protective barrier and the heterogeneous medium of plant secretory tissue to increase the stability of EO.

Encapsulation: Biomimetic carrier technology based on the protection of independent micro/nano-unitization.

Biomimetic technology based on independent micro/nano-unitization is a construction of micro- or nano-units that simulates the ground tissues of plants to encapsulate EO droplets via the cell wall or outer shell independently, and is widely known as encapsulation technology. Encapsulation technology involves the bonding and entanglement of polymer molecules, constructing dense aggregates as biomimetic polymeric carriers, and can occur in various form such as microcapsules, nanocapsules, microspheres, nanoparticles, and micelles [29] (pp. 301–306, 315–317). Commonly, the EO will be scattered and trapped by the polymeric aggregations due to interactions between polymer molecules, so that the oil droplets can be sheltered by the shell or matrix of the biomimetic polymeric carrier during the encapsulation process. Under micromorphology, no matter in micro- or nano-scale, encapsulation carrier systems often show numerous small sacs or spherical bodies with tiny holes on their surface.

In addition to hydroxyl groups, the repeating unit of a polysaccharide has a six-membered ring structure with strong intramolecular and intermolecular forces conducive to molecular cohesion to form an encapsulated carrier. In conclusion, EO is individually enclosed by encapsulated carriers and plant ground tissue cells using the polymeric wall or polymeric matrix and cell protective structures like the cell wall and cell membrane, respectively. The encapsulated carrier is a biomimetic structure that corresponds to an individual plant cell to increase the stability of the EO.

Solidification: Biomimetic technology based on multiprotection effects.

Biomimetic technology based on multiprotection effects involves further cross-linking of the intermediate dosage form to strengthen the restrictions between EO droplets and the polymeric carrier based on the simulation of internal secretory tissues, which are specialized structures in plant cells that store EO. The procedure often starts with encapsulation and/or emulsification to prepare the intermediate carrier, followed by solidification, especially via processes like gelling and filming. It can be considered optimization of the carrier system via solidification, leading to the formation of multi-polymeric structures that present in a semi-solid or solid form as a final carrier. The carriers in semi-solid or solid form correspond to the interior secretory cells, which are highly specialized to protect the EO and are generally named composite carriers. In terms of pharmaceutical preparations, biomimetic drug delivery systems are normally categorized by gel and film, that is, the EO intermediate carriers are further fixed by the polysaccharides with gelling or filming ability, and the oil droplets are primally protected by the intermediate carrier structures and additionally protected by solid structures. Hence, EO intermediate carriers are distributed in networks or matrices of other polymers in terms of micromorphology.

Indeed, the crystal lattices of polysaccharides tend to form a three-dimensional network structure via cross-linking to chemically and physically anchoring the EO within the grid mesh [30,31]. To summarize, the complex carrier and the plant’s internal secretory tissue protect the EO by means of optimized multi-polymeric structures and a specialized oil chamber, respectively. In addition, this protection is strengthened by the solid body and adjacent tissue or substance. The complex carrier primarily employs biomimicry in the multiprotection of EO within the plant’s internal secretory tissue.

An overview of the biomimetic drug delivery systems that are commonly found in EO preparation is given in Table 1.

## 3. Applications of Polysaccharide-Based Drug Carriers in Stabilization of Essential Oil

Polysaccharides can improve the stability of EOs through different plant biomimetic carrier technologies, including volatility, oxidation, photosensitivity, heat sensitivity, humidity sensitivity, and impurity (such as pH and metal ion) sensitivity. The different structures of polysaccharide materials advance through different techniques and form different types of carriers. For example, gum arabic is a good O/W emulsifying agent due to its hydrophilic carbohydrate chains with numerous hydroxyl groups and the hydrophobic amino acid residues, such as proline, in the glycoprotein [32,33]; starch, cellulose and chitosan have high crystallinity and also vary in their derivatives, allowing for the formation of encapsulation carrier systems with good performance [34,35,36]; sodium alginate, pectin, and pullulan have good water solubility, sodium alginate and pectin have gel properties, and the latter has better elasticity, making them suitable substrates to promote the formation of semi-finished products [37,38,39].

### 3.1. Applications of Polysaccharide Materials in Essential Oil Biomimetic Drug Delivery Systems Based on Emulsification

#### 3.1.1. Gum Arabic

Gum arabic contains 70% polysaccharides, 2% proteins, and cations such as calcium, magnesium, and potassium. There are various monosaccharides in gum arabic, providing both hydrophilic and hydrophobic groups. Hence, gum arabic is effective in its amphiphilicity and is mainly used as a natural O/W emulsifying agent. Through emulsification, gum arabic may develop a smoother EO in vitro release curve within 25 h than cyclodextrin inclusion, while also guaranteeing a smaller particle size, higher yield, and higher encapsulation efficiency (EE%) [40]. Additionally, compared with other emulsifying agents like tween 80 and lecithin, a gum arabic-based emulsion system could have minimal zeta-potential change after loading EO. However, its particle size change in 28 days was the highest, showing that its emulsifying ability was weaker than that of tween 80 and lecithin under an emulsifier/oil mass of 0.5 [41]. Hence, it is always suggested that it is used with other emulsifying agents such as pectin, sodium alginate, gelatin, etc. [29] (pp. 104–105). Moreover, hydrophobic Nile red is used to label the contents of milk droplets, and it has shown directly that gum arabic forms a well-sealed interface film layer, which further supports the statement that the gum arabic emulsion system may provide a strong enough biomimetic protective effect of heterogenous dispersion on EO [42].

Recently, gum arabic has been modified to slow down the digestion rate of oil in emulsion. The esterification of gum arabic with octenyl succinic anhydride (OSA) is preferred instead of aging modification and protein modification. The ability of OSA gum arabic to be emulsified in terms of stabilization is as high as of 20 wt% oil at a low concentration of 0.6 wt%, and breakage of the emulsions did not occur in 40 days of storage in 25–90 °C [43]. When OSA gum arabic was applied to cinnamon oil, the results showed that the particle size of the cinnamon oil emulsion was effectively reduced by OSA gum arabic, and the particle size of the emulsion droplets basically remained the same during storage, showing that it has great potential and that it is feasible to prepare a stable EO emulsion using OSA gum arabic [44].

According to a recent study [45], in the determination of structural changes of amphiphilic substances of gum arabic at the oil–water interface, it was stated that the emulsifying property of gum arabic was related to the electrostatic repulsion of glucuronic acid against highly branched molecule chains. Additionally, it should also be noted that the composition of proteins in gum arabic may also significantly affect its amphiphilicity and determine its emulsifying ability. Hence, the characterization of zeta-potential, particle size, polymer dispersity index, and other parameters is important for indicating the quality of EO–gum arabic-based emulsion systems. By all accounts, the characteristics of an emulsification drug delivery system based on gum arabic are simple, that is, only simple mixing is needed as the preparation method and the composition is simple, including EO, gum arabic, and water. Generally, gum arabic can emulsify 10–20 wt% of EO, but the regularity and pattern of its protection of volatile components of EO need to be further elucidated.

#### 3.1.2. Others

Gums such as xanthan gum, tragacanth, and guar gum are polysaccharides similar to gum arabic in terms of their chemical properties. Traditionally, they play the role of O/W emulsifying agents in drug delivery systems of EOs. Their emulsifying processes are basically same as that of gum arabic, that is, they form an interface film layer due to the hydrophobic and hydrophilic functional groups in their molecular structures. However, their emulsifying abilities are commonly weaker than that of gum arabic due to the absence of amphiphilic protein. According to our search, we found that the gums stated above tend to be used with synthetic emulsifiers to disperse the oil–water system [46]. Since their application as the main emulsifying agents in EO drug delivery systems is thought to be limited, they are not discussed in detail, but from another perspective, these gums are able to form gel-based or film-based delivery systems of EO. We will discuss this in 3.3.4.

### 3.2. Applications of Polysaccharide Materials in Essential Oil Biomimetic Drug Delivery Systems Based on Encapsulation

#### 3.2.1. Starch

Starch is made up of maltobiose. In its preparation, gelatinization is often required to promote the dissolution of starch molecular chains, and hence, the chains stretch and tangle to form a networking hydrocolloid carrier system. Recently, EO loading capacity (LC%) is generally more than 15%, while the EE% is above 70%. Similar to gum arabic, esterified OSA starch with better hydrophobicity is frequently used to ensure an adequate LC% of EO. For ginger EO, the protection by OSA starch by its microcapsule, which was determined using a general heat stability test for 10 days, could be increased by 40% [47]. On the other hand, when an intact yeast cell wall was used as the wall material to encapsulate ginger oil, the stability of the carrier in a baking experiment was only increased by 9%, showing that the biomimetic level of OSA starch is probably better than that of the cell walls of microorganisms [48].

In addition, porous starch, another example of modified starch, can be used as a new adsorber for the solidification of EO instead of traditional cyclodextrin. Porous starch with a hollow structure can adsorb small molecules of EO in its large surface area via van der Waals forces. As simple as the self-assembly of emulsification, only a mixing process is needed for the absorption of EO by porous starch, and recently, it was found that EO-loaded powder can be directly compressed into tablets. Further, the aroma map of volatile compounds in the obtained powder and tablet became significantly smoother, demonstrating another benefit of starch in improving the stability of EO from the perspective of production [49]. Additionally, it was found that the EO absorbed by porous starch can be further processed into microcapsules by combining them with other polysaccharides as wall materials. The time required for the complete release of EO could be prolonged by eight times, proving that the involvement of porous starch was conducive to optimizing EO stability in an encapsulated carrier system [50].

In fact, the mechanism of starch in improving EO stability is related to the helix chain of starch, and there are a lot of hydrophilic hydroxyl groups on the outer surface of the helix chain. Thus, the inner of helix chains are hydrophobic channels, allowing the entry of small molecules of EO. Recently, it was reported that the entrance process is caused by the presence of EO molecules, which alter the conformational isomerism of starch, trapping the EO molecules between the helices. The starch-based carriers are then entangled and the EO molecules are spontaneously encapsulated in the molecules, which gives the starch carrier system the advantages of needing fewer excipients and simple preparation.

#### 3.2.2. Cellulose

Cellulose is made up of cellobiose. Since its repeating unit is a rigid molecule that is water-insoluble, modified cellulose is needed and mainly used in EO preparation (for example, ethyl cellulose). In the encapsulation of EO, ethyl cellulose was used to construct a polymeric matrix, and the result showed that the UV absorbance of EO basically remained unchanged after 90 days of storage. Additionally, a photostability test showed that cellulose reduce the photosensitivity of EO by half, enhancing EO’s resistance to the light exposure with the help of cellulose [51]. As a composite wall material, cellulose nanocrystal was used. Two types of lemon fragrance microcapsules were prepared using chitosan-sodium tripolyphosphate and chitosan-cellulose nanocrystals, respectively, through the ion cross-linking method, and the result of thermogravimetric analysis (TGA) showed that the weight loss of the latter was always smaller than the former in the water evaporation and flavor escape stages. While in the weightless stage of the thermal degradation of wall material, the maximum degradation temperature of the latter is higher than that of the former, indicating that cellulose forms a thicker carrier wall, which is comparable in thickness to the walls of plant cells like parenchymal cells and even cork cells, so that cellulose is often found to be more effective in preventing instabilities caused by heat, etc. [52].

In the last 20 years, another source of cellulose, bacterial cellulose, has been developed. Its chemical structures are the same as those of plant cellulose, but it is purer, more designable, and more controllable, has better physiochemical properties, and is also definitely safe [53]. Although there are no reports on the application of bacterial cellulose-based EO delivery systems, bacterial cellulose nanocrystal has been used as stabilizer to prepare hydrophobic alfacalcidol Pickering emulsion. It was found that the particle size of emulsion droplets decreased as the cellulose concentration increased, indicating that bacterial cellulose can improve the stability of hydrophobic drugs through emulsification [54]. Hence, it is feasible to stabilize EO via carrier technology.

In fact, as the main composition of the plant cell wall, cellulose highly mimics the encapsulation of EOs by the plant’s tissue structure; thus, it can have an LC% of 35–88% but an EE% ranging from 13–74% due to EO compatibility and other factors such as the preparation method [51,55]. Studies [56,57] have shown that after the encapsulation of EOs, the release of EOs no longer depends on their own vapor pressure and boiling points, but relies on the nature of the encapsulation material, that is, cellulose, which has its own stability and is easier to cross-link due to its rigid molecular structure. Therefore, the carrier has strong stability against environmental influences such as heat, light, and humidity, and there are many reports on its combination with other materials in the stabilization of EO.

#### 3.2.3. Chitosan

Chitosan is made up of chitobiose. It is mainly derived from animal shells but is as rigid and stable as plant cellulose. The protective effects of chitosan towards EO were investigated, and the TGA results showed that the weight loss of oil-loaded chitosan nanoparticles was below 20%, while that of EO was around 90% [58]. The difference between chitosan and cellulose is an amino group at the C_2_ position. The amino group, which can be protonated under weak acidic conditions, will carry a positive charge, resulting in different properties of the EO delivery system. For example:PH responsiveness: The reversible cationization makes the EO carriers tolerant to pH cycles [59]. The direction of the reaction and the degree of cationization will change as the pH condition changes, and the release behavior of the EO carriers varies significantly at different pH levels [60].Bioadhesion: Since chitosan can be protonated in acidic conditions, it can adhere to the surface of various negatively charged cell membranes such as red blood cells and mucosal cells. Then, it can swell to form an in vivo–in situ smart gel [61], resulting in a hemostasis effect, mucoadhesion, targeted drug delivery, and other drug properties. It can also capture bacteria through this electrostatic attraction and synergistically improve the microorganism inhibition of EO.

Recently, derivatives of chitosan have been needed to satisfy the demands of various drug delivery properties. Among them, the modification of trimethyl chitosan has been the most popular. When N,N,N-trimethyl chitosan was used to prepare *Ocimum gratissimum* EO nanoparticles, the LC% and EE% of the nanoparticles were increased by about 10%, while the cumulative release in 24 h under different pH conditions was also increased by 10–20%, compared with unmodified chitosan-based nanoparticles, showing that N,N,N-trimethyl chitosan is conducive to the formation of a highly mimicked system [62]. Moreover, chitosan, oligochitosan, trimethyl chitosan, and carboxymethyl chitosan were, respectively used as raw material to form new carrier system of hydrophobic curcumin liposomes. It was proven that the trimethyl chitosan carrier system had the strongest corneal permeability, demonstrating that chitosan EO preparations have great potential in corneal drug delivery as well as mucosal drug delivery [63].

Chitosan is the only natural basic amino polysaccharide [64]. Its EO drug delivery systems recently showed LC% values of 6–70% and EE% values of 29–90% [58,60,61,62,65]. As a result of the high LC% and EE%, we deduce that the positive charge of chitosan probably mimics the electrical property of the plant cell membrane to a certain extent. There is a lack of in-depth research on the EO encapsulation mechanism of chitosan, but it is mainly believed that its foundation is closely related to the properties of its amino group. Generally, it is still emphasized that the presence of amino cations plays the main role in EO encapsulation. By adjusting the pH of the system, EO drug delivery systems based on chitosan can be characterized by resistance to environmental factors such as pH and ions. Moreover, the positive charge allows for the designation of a targeted drug delivery system and transdermal drug delivery system due to its influence on cell permeability, etc. [66].

#### 3.2.4. Others

Inulin is another polysaccharide that has improved EO stability through the encapsulation technique. However, a study investigated EO stability improvement by OSA starch, inulin, chitosan, and other wall materials in the preparation of microalgal oil microcapsules, and found that the regularity of inulin as the main carrier in improving the stability of EO was poor [67]. Additionally, inulin has rarely been used in EO delivery systems in previous years, so it will not be discussed in depth.

### 3.3. Applications of Polysaccharide Materials in Essential Oil Biomimetic Drug Delivery Systems Based on Solidification

#### 3.3.1. Sodium Alginate

Sodium alginate is made up of M units (β-D-mannuronic acid) and G units (α-L-guluronic acid) in the form of triblock copolymers with GM, GG, and MM repeating units. In the presence of divalent cations, sodium ions from the G units will be displaced to form a calcium alginate gel network, leading to the stacking of G units and the formation of an egg-shaped molecular structure known as the egg box model. The three-dimensional gel network structure of the egg box model will take on the role of EO encapsulation during the forming stage [68]. Instead of general gelling and filming agents [69], the special gelation mechanism of sodium alginate is currently used to develop injectable gels [70]. Eucalyptus EO cyclodextrin inclusion powder was redissolved and swollen in sodium alginate solution, and then, CaCl_2_ solution was quickly injected after tracheal intubation. Hence, an ion-sensitive in situ gel could be formed in the lung and shown via in vivo fluorescence imaging. The full dissolution time of the EO-gel delivery system was more than 12 h, indicating that the sodium alginate could effectively provide slow release of the drug in the lesion through the multiprotection of powdering, follow up by in situ gelation [71].

As a sodium salt, sodium alginate is hydrophilic and so limits LE% for hydrophobic drugs like EOs. There have been many studies on the modification of sodium alginate, but no drug delivery systems that use EOs as raw materials and are based on modified sodium alginate have been investigated. Positively, acylated sodium alginate loaded with hydrophobic emodin was proven to increase LE% by 4–10% and slow down 36 h cumulative drug release by 6–15%, showing the potential of sodium alginate for the biomimetic protection of hydrophobic EOs [72].

In fact, as natural polymers, blocks of sodium alginate molecules are randomly interweaved in different proportions; hence the crystalline regions and three-dimensional structures are not stable at all. Inter- and intra-batch variability may disadvantage the stable production of related drug delivery systems. As a result, the LE% ranges from 9 to 36%, and the EE% ranges from 57 to 95% [73,74,75,76]. However, it is interesting to find that the COONa groups could cause acidity in sodium alginate, which contrasts with chitosan. Therefore, sodium alginate, which is negatively charged, is often used with chitosan to prepare a dense encapsulation drug delivery system through the combination of complex coacervates and interfacial interaction methods. The complex carrier’s wall structure will include an inner layer of polysaccharide, middle layer of polyelectrolyte film (formed through the attraction of the opposite charges of chitosan and sodium alginate), and outer layer of another polysaccharide [73,77,78,79].

#### 3.3.2. Pectin

Pectin is made up of three kinds of galacturonic acid as repeating units, and it possesses various functional groups with different chemical properties. For instance, pectin is amphiphilic since it has acetyl, ferulic acid ester, carboxyl, and other groups, and therefore, it is an emulsifying agent [80]. Additionally, pectin has many ions like sodium, potassium, calcium, etc., causing it to easily form ion cross-linking gel via a substitution reaction. In pharmaceutical excipients, it can be divided into high-methoxy (HM) and low-methoxy (LM) pectin, which differ in their gelation properties. HM pectin normally forms a gel through non-covalent bonding in galacturonic acid regions, while LM pectin can form an egg-shaped molecular structure like calcium alginate gel, with more binding sites and stronger binding strength. Commonly, HM pectin performs better than LM pectin in emulsification systems. A rheological study found that the viscosity of an HM pectin delivery system increased as the ratio of pectin/emulsion increases, and showed that the multiprotection of pectin firstly forms an emulsion, and then, becomes a semi-solid gel due to the increase in networking degree [81]. Meanwhile, in emulsification, microcapsule powder was added into a pectin and gelatin solution, respectively, to compare their gelling ability. It was found that pectin increased the hardness of the microcapsule powder by 30%, which is twice that of gelatin, and could release EO for at least 3h [82].

The physical strength of highly esterified natural pectin gel is not as good as that of LM pectin; hence, the research on pectin modification focuses on the reduction in the methyl esterification degree and improvement of the water absorption property [83]. However, esterification has allowed pectin to be more compatible with hydrophobic EOs. The chemical proportions of EOs are basically the same before and after their preparation [84]. HM pectin can also improve the thermal stability and pharmacological activity of EO multiple times [85]. Therefore, compared with LM pectin, HM pectin is generally used in carrier systems to improve the stability of EOs, while other modified pectins are rarely used.

In fact, the mechanism of pectin in improving the stability of EO is complex, and its application in EO is rare. According to the available data, the LE% is only 4%, and the EE% is only 25%. However, recent studies have shown that the mechanism of pectin in improving the stability of EOs is related to the molecular weight of pectin, the degree of esterification of pectin, the galacturonic acid cross-links between pectin chains, and the interactions between pectin and other molecules (such as EO and emulsifying agent molecules) [86]. In addition, there are various types of EO and pectin. For example, cinnamon oil may be different in chemical composition since it can be extracted from different parts of plants, such as the inner bark and leaves, while pectin can be derived from different medicinal plants [87,88]. So, it should be noted that the protective effect is easily affected by plant source, preparation method, etc., leading to variation in drug carriers. Usually, EO drug delivery systems based on pectin are in the form of emugels [89]. The complex carrier highly mimics the multi-protection of plant EOs. This dosage form has the advantage of simplifying the formulation and process method, and the EO proportion is similar before and after the preparation due to the high degree of esterification. This is beneficial to the internal balance and function of EOs.

#### 3.3.3. Pullulan

Pullulan is a maltotriose trimer that does not have a gelling property, but it has good elasticity and solubility. Its solution is electrically neutral, and hence, it is often used as a filming agent to solidify EO emulsion or encapsulation. Clove EO nanoemulsion and Pickering emulsion were prepared and, respectively, added into a matrix solution made using pullulan/gelatin to form a thin film composite. As a result, when the film solutions were kept at low temperature, room temperature, and body temperature, the release curve showed continuous growth in 72 h, indicating that the volatility of clove EO was slow and incomplete, that is, the retention rate was always maintained at a high level [90].

The modification of pullulan focuses on increasing its hydrophobicity to reduce the moisture absorption of pullulan and to promote its compatibility for hydrophobic drugs. A variety of methoxylated whey protein isolate–pullulan composite aerogels was prepared to adsorb clove EO, all of which delayed the total release rate of EO differently [91]. However, there are no reports on the application of modified pullulan thin layer composites in the stabilization of EO.

Recently, the application of pullulan in the solidification of EO intermediate carriers has been rare, and hence, there is a lack of the experimental studies that show its LE% and EE%. However, it is a fact that pullulan can further stabilize EO through plant biomimetic technologies, especially via filming, since a lot of hydroxyl groups on its surface allow pullulan to form intermolecular hydrogen bonds with cross-linking agents, proteins, and other substances to construct dense network structures of EO carriers [92].

#### 3.3.4. Others

Recently, there has been a focus on transforming the gums in Section 3.1.2 into a gelling systems to discover their full potential. There are also some other gums, such as guar gum and carrageenan, which can be used to form EO-loaded gels or composite films. However, compared with pectin and sodium alginate, the applications of others gum are fewer (and there are even no reports for some gums), and they may always need to be used with other polysaccharides such as pullulan, chitosan, etc. There are not enough supporting data to review their protective effects in EO drug delivery systems. Positively, we believe that the gums in polysaccharides may play a role in gelling or filming delivery systems of EO based on recent studies, but their effectiveness in EO stabilization via biomimetic drug delivery systems may be limited at present.

The applications of polysaccharides in EO biomimetic drug delivery systems, along with their stabilization evaluation, are listed in Table 2.

## 4. Abilities and Potential of Polysaccharides in Stabilization of Essential Oil

Polysaccharides show various properties based on different functional groups in the drug delivery of EOs via polysaccharide materials. For example, gum arabic, which is amphiphilic, can attract water molecules using hydrophilic groups and surround oil molecules using hydrophobic groups. Thus, an interfacial film will form, while the main application of gum arabic is as an emulsifier of microemulsion, nanoemulsion, and Pickering emulsion. Further, gum arabic can encapsulate EOs via a complex coacervation method, with B-type gelatin used as a composite wall material. The encapsulation process is based on the neutralization of basic gelatin and acidic gum arabic at pH 4–5 to precipitate the microcapsules. Secondly, starch is always the first choice for encapsulation since it is cheap and safe. In general, the preparation process includes degreasing of the starch (OSA), gelatinizing the starch at a high temperature, and the mixing of EOs with the starch, followed by cooling, centrifugation, washing, and drying. Thirdly, the interaction between the molecular chains of cellulose is strong due to its high rigidity, and EO carriers are more stable. The EOs encapsulated by cellulose may be more resistant to heat, light, humidity, and other external energy; hence, it is suggested to be used to slow and control the release of EO preparations.

Chitosan is an important material for pH-sensitive and targeted preparations since its reversible protonation reaction allows carrier systems to be more resistant to freeze–thaw cycles and pH cycles. Additionally, chitosan is another ideal matrix for drug targeting microspheres because of its hemostatic, antibacterial, and other pharmacological properties. In contrast, sodium alginate, which carries an opposite charge, is also pH-sensitive. Its solubility is pH-dependent, while its ionic sensitivity allows for the formation of a hydrogel, which is even injectable in situ. Consequently, sodium alginate is suitable for stomach floating preparations, and conducive to the colonic release of oral preparations.

Moreover, pectin is characterized by a complex chemical composition and highly branched structure. The structures of pectin are advantageous in the formulation of gel networks, films, and emulsions. Finally, the linear polymer pullulan is conformationally stable, so its drug carrier is strong in antioxidant capacity, heat resistance, ionic strength resistance, pH resistance, UV resistance, and water solubility. Additionally, it is a kind of Newtonian fluid with advantages for film generation, high plasticity, and lubrication.

An overview of the characteristics of polysaccharides stated above is shown in Table 3.

In fact, material combination is the main way to optimize the performance of EO drug delivery systems. Among the polysaccharides stated, chitosan and sodium alginate are the most promising combination. The composite carrier can be prepared by equally mixing drug-loaded solutions with opposite charges, or by a mixing a drug-loaded solution with another polysaccharide-based solution. Hence, the EO droplets are respectively protected by either a dense polyelectrolyte layer or by aggregation formed via attraction between opposite charges. Anbazhagan et al. [100] prepared a tea tree EO–chitosan nanoemulsion, and slowly added it to sodium alginate solution, follow by dropping, stirring, and ultrasonic dispersion; then, tea tree EO-chitosan–sodium alginate microspheres were formed. Its LE% is up to 15% and its EE% is 71%. TGA showed that the weight loss of the carrier system was only 75% at 800 °C, indicating that the EO carriers were physically strong enough to protect against heat decomposition, which effectively improved the thermal stability of the tea tree EO. In addition, Lu [101] used chitosan–sodium alginate to prepare gastric-acid-sensitive-reagent-loaded peptides, while Mei [102] used chitosan–sodium alginate–pectin to prepare oral-intake colon-targeted microspheres, both of which could effectively be release in different pH conditions, demonstrating the gastrointestinal pH responsiveness of the two chargeable polysaccharides. Hou [103] used the tea polyphenol as a core material and sodium alginate as basic wall material, and, respectively, combined the composite wall material with gum arabic, inulin, and chitosan, which differ in their charge properties. Compared to gum arabic and inulin, the chitosan–sodium alginate carrier system could retain the most of volatile tea polyphenols. Although this kind of preparation design was not directly applied to EO in the study, the self-assembly and strong stability of the combined polysaccharide carrier structure is still feasible in the stabilization of EOs due to its strong electrostatic forces. In brief, this combination has advantages in the formation of gastric floating preparations, slow or modified release preparations, pH and mucosal targeted preparations, etc., and is especially recommended in the innovative new drug development of traditional Chinese medicine with the bidirectional regulation property of gastrointestinal administration.

Secondly, both cellulose and pectin act as cell-wall components in plants, but in biomimetic carrier systems, cellulose can enhance the gelation of pectin and promote the formation of colloids, mimicking the storage of EO in colloidal conditions like the cytoplasmic matrix. However, the application of polysaccharides, especially pectin, in EO drug delivery systems is limited at present since they are mainly used as antimicrobial agents in the field of food preservation [104,105]. Interestingly, Qi et al. [106] encapsulated a chitosan–sodium alginate microgel with ethyl cellulose, which made the volatile hydrophobic drug methyl blue release in vitro for up to 15h, and it was certainly pH-sensitive. Meanwhile, the study also showed that the emulsion-loaded sodium alginate and chitosan were highly porous networks, while cellulose pelleted the carriers, which further enhanced the drug encapsulation protection of the EO carrier system.

## 5. Discussion

A review [107] classified the properties of the microscopic structure of EO in herbs and found that the chemical composition of EO probably correlates with its natural form in plants. According to this study, the chemical composition of EO that is protected by plant secretory tissues is dominated by a single component (40–80%), and the high proportion of the main volatile component is unstable. The chemical composition of EO that is protected by basic tissue is mostly 20% of a certain component; if the content of the main volatile components is lower, then the EO is more stable. The majority of plant EOs have no specific structural form (they do not exist in secretory tissue or basic tissue) and commonly contain a high proportion of fatty acids, while the content of other components is less than 10%. The chemical composition of most EOs is varied and dispersed. The proportion of main volatile components is the lowest and most stable. At present, the view that EOs exist in different structures in plants and may affect physical and chemical properties such as volatility needs further study. However, in support of this view, EOs extracted from plants with secretory tissues are frequently used in preparations, for example, cinnamon oil, clove oil, patchouli oil, zedoary turmeric oil, etc. This may be related to the dominant chemical compositions that are conducive to the evaluation of preparations.

Recently, in the development of EO drug delivery systems, polysaccharides have received attention since they are nontoxic, biocompatible, and absorbable. However, as a natural substance, the stability of polysaccharides is weaker than that of synthetic polymers, and their physical and chemical properties cannot sacrifice the needs of the industrial preparation of EO carriers. Compared with traditional cyclodextrin inclusion technology, polysaccharide carrier technology is new and leaves much to be desired. The LE% and EE% vary greatly and are not ideal, so they are still in the stage of laboratory research. Based on the above reasons, modified polysaccharides that are semi synthetic have become the key to the construction of efficient carriers. There are four main research directions for the modification of polysaccharide materials to improve the inner structure of polysaccharide particles. Firstly, the modification method involves the introduction of different functional groups such as the OSA group, the phosphate group, etc. Hence, the self-stability of polysaccharides, such as heat resistance, freeze–thaw resistance, pH resistance, solubility, and hydrophobicity, may be improved. The second is modified-porosity polysaccharides. The particle of the polysaccharide is hollowed via different techniques, such as enzymes and ultrasounds, so that the EO can be dispersed, adsorbed, and solidified by the polysaccharide. Zhang et al. [108] used porous starch to adsorb tea tree EO, chitosan, and sodium alginate, as wall materials, to prepare composite microcapsules and achieve a low EO loss rate of 7.37% in 14 days. The third is the blending modification of polysaccharides and other materials. For example, Song et al. [109] found that when chitosan and attapulgite are mixed, the retention rate of peppermint EO can be increased by 90.45%. The study shows that a simple blending modification research mode to obtain new carrier materials with complementary properties can protect EO effectively. In addition, microorganisms are another tool to modify polysaccharides. For example, as another derivative of starch, xanthan gum is applied to stabilize cinnamon oil. The results of a study showed that the presence of xanthan gum may significantly increase the thermal stability of cinnamon oil in terms of heat decomposition [110]. However, other microbial polysaccharides such as diutan gum and gellan gum have not been applied to EO stabilization to date. In summary, the application of polysaccharides as carrier materials in the stabilization of EOs occurs mainly through modification methods such as enzymatic, physical, chemical, and blending, and thus, improve the properties of polysaccharides to improve the stability of EO through the construction of high-performance and multifunctional biomimetic carriers.

Overall, polysaccharides, which are common plant cell components, simulate the protective environments of EOs in plant tissues, and so construct a biomimetic system via carrier technologies. For the EO, it is compatible with polysaccharides that mimic plant tissue. However, EOs have distinct and intricate chemical compositions, while polysaccharides are heterogeneous due to their different degrees of polymerization. Therefore, when they are mixed, the affinity between their molecules is different. The capacity and strength of a polysaccharide material compared to EOs are also different. Because of this varying compatibility, the biomimetic level of EO drug delivery systems varies. Therefore, it can be concluded that one of the keys to the success of EO protection by polysaccharide carrier systems is determining the compatibility of EO with polysaccharides. At present, there is no research on the compatibility of polysaccharides with EO. A single-factor experimental design can be considered in an EO preparation study to specifically investigate the most suitable carrier material for a particular EO. Sun [65] encapsulated clove oil with sodium alginate, cellulose, gum arabic, and other wall materials, respectively, and the results showed differences in the antioxidant rate and in vitro dissolution rate. Hence, their study proved the difference in compatibility between polysaccharides and EO using the research mode of “same core different shell”. Inversely, the study used gelatin–chitosan as the wall material to encapsulate Angelica oil and clove oil, respectively, and prepared two microcapsules with the “same shell different core” using the same parameters. It was found that their stability was also different but could be easily improved via an orthogonal experiment. Obviously, the research modes of “same core different shell” and “same shell different core” are recommended for the study of EO drug delivery systems since they may explain the compatibility of polysaccharides and EOs. Moreover, they may optimize the formulations effectively. Meanwhile, in basic pre-formulation studies, the relationship between the naturally existing form of EO and its stability, the synthesis of polysaccharides in plants, and the synthesis and storage of EO in plants should be investigated, and efforts should be made to improve the biomimetic level of polysaccharide carrier systems.

## Figures and Tables

**Figure 1 polymers-15-03338-f001:**
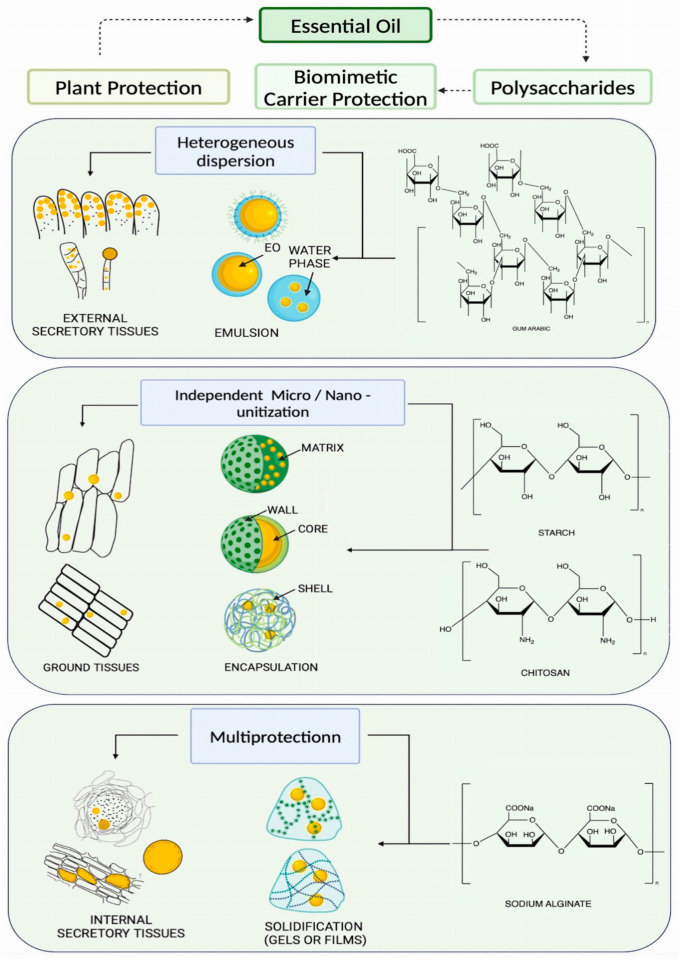
Biomimetic drug delivery system based on polysaccharides of EO.

**Table 1 polymers-15-03338-t001:** Polysaccharide-based EO carrier technologies.

Classification	Dosage Form	Polysaccharides Available	Foundation of EO Stabilization	Pros	Cons
Emulsification	Microemulsion	Gum arabic, pectin	The EO is isolated in a core stabilized by polymeric interfacial film and heterogenous medium.	Self-assembly EO used directly as oil phase Easy to solidify	Weak storage stability Relies on traditional surfactants
	Nanoemulsion			EO used directly as oil phase Easy to solidify	Thermodynamically instable Unable to self-assemble
	Pickering emulsion			Self-assembly EO used directly as oil phase Easy to solidify Safe stabilizer	Weak storage stability
Encapsulation	Microcapsule	Gum arabic, starch, cellulose, chitosan, sodium alginate, pectin, pullulan	The polysaccharides build a core-shell structure to restrict the oil droplets’ movement via a rigid wall or matrix trapping	Designable and rich in form Strong targeting Easy to solidify	Relies on toxic cross-linking agents, organic solvents, etc. Unstable in vivo
	Nanocapsule				
	Microsphere		EO is distributed on the surface of the polymeric carrier and sheltered in it		
	Nanoparticle				
	Micelle				
Solidification	Gel	Sodium alginate, pectin, pullulan	EO is further fixed on the polymeric network since the polysaccharides increase the solidity of the intermediate carrier	Easy to store Strong protective ability	Low LC%
	Film				

**Table 2 polymers-15-03338-t002:** The application of EO biomimetic drug delivery systems.

Polysaccharides	Essential Oil	Dosage Form	Stability	References
Gum Arabic	Cinnamon Oil	Microemulsion	Particle size and PDI of microemulsion prepared with whey protein were basically unchanged within 7 days at 21 °C.	[93]
Gum Arabic	Du Liang Fang Essential Oil	Microemulsion Lyophilized Powder	Ligustilide content was basically unchanged within 1 month of accelerated stability test and 3 months of long-term stability test. Ligustilide content was 2.7 times that of EO in high-temperature test and 6 times that of EO in light exposure test.	[94]
Gum Arabic	Angelica Essential Oil	Microcapsule	Cumulative release was 1.5 times lower than that of EO Ligustilide and atractylodes contents were 1.25 and 1.38 times that of EO in high-temperature test, and 1.75 and 1.77 times that of EO in high-humidity test.	[95]
Gum Arabic	Pepper Oil	Microcapsule	In vitro release lasted for 6 h, and the endpoint was at 80%. Weight loss was 7.25 times less than that of EO at 200 °C in TGA. The retention rate always exceeded 50% at pH 2–8, and exceeded 70% at ionic strength of 0–320 nM.	[96]
Gum Arabic	Thyme Essential Oil	Microemulsion-Microcapsule	In vitro release lasted for 24 h and the cumulative release endpoint of gum arabic–cyclodextrin was about 5% lower than that of cyclodextrin.	[40]
OSA Gum Arabic	Cinnamon Oil	Microemulsion	Particle size and dispersion index (PDI) were basically unchanged within 7 days at 4 °C, and the emulsion was not broken.	[44]
OSA Starch	Ginger Oil	Microcapsule	Changes in iodine value and peroxide value were 7 and 5 times less than EO in high-temperature test, 2 and 5 times less than EO in high-humidity test, and 7.5 and 2.6 times less than EO in light exposure test.	[47]
Porous Starch	Tangerine Peel Oil	Absorber	Volatilization of limonene lasted for 36 h. The decomposition peak reached around 100 °C in TGA.	[49]
Corn starch	Garlic Essential Oil	Self-Assembler (Mixture of Dried Powder)	DADS content was 2 times that of EO after storage at 50 °C for 12 days. Weight loss was 5–10 times less than that of EO at 200 °C in TGA.	[97]
Potato Starch				
OSA Starch	Amomum Tsaoko Essential Oil	Microcapsule	Weight loss was 16 times less than that of EO at 200 °C in TGA.	[98]
Ethyl Cellulose	Babchi Essential Oil	Porous Microspheres	The total drug content was basically unchanged after storage at 40 °C and 75% humidity for 90 days. Differential scanning calorimetry (DSC) curve showed that the heat release peak was lower and blunter than that of EO. Change in absorbance was 1.75 times less than that of EO after light exposure for 1 h.	[51]
Sodium Cellulose Starch	Lemongrass Essential Oil	Microspheres	Weight loss was only 2% within 250 °C in TGA.	[55]
Chitosan	Lemongrass Essential Oil	Nanoparticle	In vitro release curve growth over 60 days and endpoint below 50% Weight loss was 7.2 times less than that of EO at 200 °C in TGA.	[58]
Chitosan	Clove Oil	Pickering Emulsion	The breaking of emulsion was delayed as chitosan concentration increased. Same appearance after storage at 25–55 °C for 48 h. No oil leakage when pH under 6. Same antibacterial effect after 5 pH cycles.	[59]
Chitosan	Magnolia Essential Oil	Microspheres	Volatilization was 12 times less than that in EO.	[61]
Chitosan	*Ocimum gratissimum* Essential Oil	Nanoparticle	In vitro release lasted for 24 h, and the endpoint was below 80%.	[62]
Chitosan	Angelica Essential Oil	Microcapsule	In vitro dissolution for 1 h was 9 times less than that of EO.	[65]
Chitosan	Clove Oil		In vitro dissolution for 1 h was 2 times less than that of EO.	
Chitosan + Sodium Alginate	Atractylodes rhizome Essential Oil	Microcapsule	Simulated release curve of gastric juice and intestinal juice showed that EO is pH-resistant. The EO decomposition peak reached around 400 °C in TGA.	[73]
Sodium Alginate			DSC curve showed that the EO decomposition peak reached 420 °C.	
Sodium Alginate	Eucalyptus Essential Oil	Inclusion-Gel	Gel dissolution process lasted for 12–24 h in vivo.	[71]
Sodium Alginate	Cinnamon oil	Adsorber-Gel Microspheres	In vitro releases were significantly slower than those of EO at different pH levels. Weight loss was 3 times less than that of EO at 200 °C in TGA.	[74]
Sodium Alginate	Nutmeg Essential oil	Microcapsule	Centrifugal stability index increased as sodium alginate concentration increased.	[75]
Sodium Alginate	*Perilla frutescens* (L.) *Britt.* Essential Oil	Microcapsule	In vitro release lasted for 24 h and endpoint was below 80%. Weight loss was 5 times less than that of EO at 200 °C in TGA.	[76]
Sodium Alginate	Osmanthus Essential Oil	Nanocapsule	The weight loss temperature of EO was doubled in TGA. In vitro release lasted for 2 h at 105 °C and the final weight loss was 18 times less than that of EO.	[77]
Sodium Alginate	Thyme Oil	Composite Microcapsule	In vitro release lasted for 60 days and endpoint was 2–4 times less than that of EO. The cumulative release of 2-, 4-, and 6-layer composite microcapsule samples was about 2, 3.5, and 5.2 times less than that of EO after 5 h heating.	[78]
Sodium Alginate	*Coriandrum sativum* L. Essential Oil	Microcapsule	In vitro release lasted for 4.5 h and endpoint was below 80%.	[99]
HM Pectin	Orange Oil	Nanocapsule	TSI of high-concentration pectin was basically unchanged after storage for 18 days.	[81]
HM Pectin	Pink Pepper Oil	Microcapsule	The drug contents were, on average, half those of EO after storage for 20 days.	[84]
HM Pectin	Jasmine Oil	Nanoparticle	Thermal stability was 1.64 times that of EO in TGA.	[85]
HM Pectin	Lemon Oil	Emugel	Turbiscan Stability Index (TSI) was basically unchanged after storage for 15 days.	[89]
Pullulan	Clove Oil	Nanoemulsion-Composite Film	In vitro release curve growth over 72 h and endpoint was below 70%.	[90]
		Pickering Emulsion- Composite Film		
Pullulan	Licorice Essential Oil	Double-layer Microcapsule	The water vapor permeability of the EO-loaded microcapsule film was 0.7 g/m^2^/days less than that of blank microcapsule film.	[31]

**Table 3 polymers-15-03338-t003:** Characteristics of polysaccharides used in EO carriers.

Polysaccharides	Main Application	Pros	Cons
Gum Arabic	Amplificant Wall material	Cheap	Poor Poor transparency
Starch	Wall material	Various kinds Cheap palatability	Hydrolysable Poor mechanical resistance
Cellulose	Wall material Filming material	Various types Cheap Strongly biomimetic, slow release	Poor solubility Acid hydrolysis
Chitosan	Wall material Microsphere matrix Filming material	Hemostatic and antibacterial activity (oral, nasal, and gastrointestinal), targeting of mucosa and cells	Expensive Animal origin Poor mechanical resistance
Sodium Alginate	Wall material Injectable gel Gastric flotation agent	Gel condition mild Powder fluidity	Humidity sensitivity Strong mechanical resistance
Pectin	Gelling agent Filming material	Strongly biomimetic, pharmaceutical ingredients before and after the same preparations	High viscosity, poor sustained release
Pullulan	Filming material	Microbial source, controllable Skin adhesion	Weak mechanical properties Little research, lack of data support in application of EO drug delivery

## Data Availability

Not applicable.

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
