# Peer review of "Stabilization of Essential Oil: Polysaccharide-Based Drug Delivery System with Plant-like Structure Based on Biomimetic Concept"

_polymers, 2023, doi:10.3390/polym15163338_

Round 1
Reviewer 1 Report
The subject matter of this review may be considered of high interest to readers in view of its potential applications in fields such as food technology or pharmaceuticals. The manuscript is well written and in almost all cases, the topics it addresses are appropriate. However, I consider that in order to be considered a reference review and to be published in Polymers, more information should be included in some aspects and some others should be revised.
The authors indicate in Table 1 gum arabic and pectins as reference polysaccharides. There are numerous current studies on the development of emulsions, nanoemulsions, emulgels and nanoemulgels with EO using other stabilizers of microbiological origin (such as advance performance xanthan gum or diutan gum). I believe that more information and references should be included so that this review can be considered as a reference in its field of study.
For the development of this type of biomimetic systems, would it be possible to use inorganic stabilizers such as fumed silica (Aerosil 200 or COK84)?
Reviewer 2 Report
The manuscript entitled “Research Progress of Drug Delivery System Based on Polysaccharides and Plant-like Structure Biomimetic Concept in Stabilization of Essential Oil” provides a review of the use of polysaccharides to stabilize essential oils (EOs) for use in drug delivery systems and discusses the biomimetic properties of EO-based drug delivery systems. This review includes a discussion on the development of polysaccharides and their derivatives, such as gum Arabic, starch, cellulose, chitosan, sodium alginate, as well as pectin and pullulan, and how these polysaccharides can be used for the development and protection of EO-based biomimetic drug carriers.
This review is fascinating and covers an important and highly relevant topic. Overall, the review is well-organized and I believe it is a valuable contribution. The writing could benefit from polishing as there are a number of issues with grammar, incorrect use of past/present tense, etc. I have provided some suggestions below, but significant proofreading will likely be required besides these suggestions.
Lines 2-4, title: The article title is a little difficult to follow and should possibly be rewritten for clarity.
Line 35: The phrase “However, it is easily loss and changing its composition after” is unclear.
Lines 38-43: The sentence “Li et al. [6] studied the process 38 of EO preparation and founded that the extraction process was maximumly leading to 39 heat transformation and accelerate the volatilization of components compared with 40 crushing, concentration and drying processes, while compared with EO, the volatile com- 41 ponents could be retained more when the raw powder is directly used in the preparation, 42 indicating that the plant cell structures was conducive to the stability of the EO” is quite long and difficult to follow. It may be helpful to break it down into 2 or 3 sentences for clarity/easier reading.
Line 50: “pulverization as a drug preparation process” can possibly be changed to “pulverization could be used as a drug preparation process”
Line 57: The phrase “causing that EO cannot exist stably in vitro for a enough storage time” is unclear.
Lines 72-73: “(except chitosan mainly from animal shells),” can be changed to “(except chitosan which is mainly derived from animal shells),”.
Line 80: “this paper classify” can be changed to “this paper classifies”.
Line 112: The phrase “to protect EO like plant tissues” is unclear.
Lines 137-139: References may be needed for the statement “There were also frequently connected to hydrophobic groups such as methyl groups, and so can be used as a surfactant to stabilize the interface while mixing of EO and water”.
Line 158: “Besides of hydroxyl groups,” can be changed to “Besides hydroxyl groups,”.
Lines 172-173: The statement “The optimized carrier is corresponding to specialized secretory cells and can be named as composite carrier generally” seems to be unclear.
Lines 192-198: May need references for the examples mentioned here.
Line 202: “Gum arabic, containing” can be changed to “Gum arabic containings”.
Line 226: “According to recent study” can be changed to “According to a recent study”.
Line 250: “another of modified starch, can” can possibly be changed to “another example of modified starch, can”.
Line 252: “Van Der Waal force” can be changed to “van der Waals forces”.
Line 271: “that water insoluble” can be changed to “that is water insoluble” or “that is insoluble in water”.
Lines 286-287: References may be needed for the phrase “By comparison, bacterial cellulose is purer, more designable, more controllable, better in physiochemical properties, and is as safe as plant cellulose”.
Lines 303-304: The phrase “It may correspond to plant cellulose in animal 303 kingdom, as rigid and stable as it.” Is unclear.
Lines 330-331: The statement “Chitosan is the only natural basic polysaccharide and with a LC% of 4-28% while an EE% of 20-90%.” Seems to be unclear and a reference should be provided.
Lines 352-353: The phrase “structure, means that a three-dimensional gel network is constructed to further encapsulation of EO, known as the egg box model” seems to be unclear.
Line 375: “is negatively charge” can be changed to “is negatively charged”.
Line 411: The phrase “there are various of EO,” is unclear, it appears that a word may be missing.
Line 433: “are no report on” can be changed to “are no reports on”.
Lines 476-477: “EO drug delivery system” can be changed to “EO drug delivery systems”.
Lines 491-493: The phrase “sore material, sodium alginate as basic wall material, and respectively combined with gum arabic, inulin and chitosam which is different in charge properties as complex wall.” Seems to be unclear.”
Line 492: “chitosam” should be changed to “chitosan”.
Line 519: “The phrase “The most of EO have no” is unclear.
While the manuscript is well-organized and covers an important topic, the writing is somewhat difficult to follow in places. Significant proofreading may be needed for the revised version of the manuscript.
Round 2
Reviewer 2 Report
Overall, I believe that the authors of the manuscript entitled “Research Progress of Drug Delivery System Based on Polysaccharides and Plant-like Structure Biomimetic Concept in Stabilization of Essential Oil” have successfully addressed the comments of the reviewers, and that the manuscript is now suitable for publication.
While I recommend acceptance, I have a couple of possible minor suggestions below.
Line 56: “causing that EO easily loss of flavor” can be changed to “causing the EO to easily lose flavor”.
Lines 455-456: The phrase “Newly, the gums in 3.1.2 are tended to form gelling system to explore more value of them” seems to be unclear.
The quality of the writing has improved significantly with the revisions. There may be a few areas for minor polishing, but it has improved.